# Agent Q: Advanced Reasoning and Learning for Autonomous AI Agents

## Abstract

Large Language Models (LLMs) have shown remarkable capabilities in natural language tasks requiring complex reasoning, yet their application in agentic, multi-step reasoning within interactive environments remains a difficult challenge. Traditional supervised pre-training on static datasets falls short in enabling autonomous agent capabilities needed to perform complex decision-making in dynamic settings like web navigation. Previous attempts to bridge this gap through supervised fine-tuning on curated expert demonstrations often suffer from compounding errors and limited exploration data, resulting in sub-optimal policy outcomes. To overcome these challenges, we propose a framework that combines guided Monte Carlo Tree Search (MCTS) search with a self-critique mechanism and iterative fine-tuning on agent interactions using an off-policy variant of the Direct Preference Optimization (DPO) algorithm. Our method allows LLM agents to learn effectively from both successful and unsuccessful trajectories, thereby improving their generalization in complex, multi-step reasoning tasks. We validate our approach in the WebShop environment, a simulated e-commerce platform—where it consistently outperforms behavior cloning and reinforced fine-tuning baseline, and **beats average human performance** when equipped with the capability to do online search. In real-world booking scenarios, our methodology boosts Llama-3 70B model's zero-shot performance from **18.6% to 81.7%** success rate (a **340% relative increase**) after a single day of data collection and further to **95.4%** with online search. We believe this represents a substantial leap forward in the capabilities of autonomous agents, paving the way for more sophisticated and reliable decision-making in real-world settings.

## 1 Introduction

The recent advances in Large Language Models (LLMs) represent a significant leap in artificial intelligence. Frontier models like ChatGPT (John Schulman et al., 2022), Gemini (Anil et al., 2023), Opus (Anthropic, 2024), and LLaMA-3 (Touvron et al., 2023) demonstrate promising reasoning capabilities that approach average human performance in a number of domains. These breakthroughs have extended the utility of LLMs from traditional chat and text-based applications to more dynamic, agentic roles, in which they do not just generate text but can take actions autonomously in a number of environments including code and software engineering (Holt et al., 2024; Zhang et al., 2024d; Jimenez et al., 2024; Yang et al., 2024), device control (Wang et al., 2024a; Zhang et al., 2024b; Chen and Li, 2024) and web applications (Hong et al., 2023; Deng et al., 2023; Zhou et al., 2024b; Lai et al., 2024a; Gur et al., 2024) among others. However, despite these advancements, significant challenges persist: LLMs still struggle to generalize effectively in interactive, multi-step environments, since they are not natively trained for such applications . This is true, even for some of the strongest models of the current generation, such as GPT-4 (Achiam et al., 2023).

A growing literature on agentic formulation seeks to address these issues; however these works mostly focus on building frameworks around prompt-based learning on existing models or limited fine-tuning on static datasets, and are thus limited by the base models' reasoning and decision making capabilities. Reasoning and planning have indeed been highlighted as core challenges for current LLMs. Since the seminal work on chain-of-thought reasoning (Wei et al., 2022), significant efforts have been made to improve these capabilities via prompt-based strategies (Kojima et al., 2022; Wang et al., 2023; Qiao et al., 2023; Yao et al., 2023a). While successful, these approaches are still bounded

by the base model's performance. Another direction of research has explored fine-tuning approaches (Zelikman et al., 2022; Pang et al., 2024), and more recently combining them with inference-time search prompting (Yao et al., 2023a) to produce fine-grained feedback. Concurrent works (Xie et al., 2024; Hwang et al., 2024; Zhang et al., 2024e; Tian et al., 2024) utilize the traces produced by search algorithms and combine them with optimization approaches (Rafailov et al., 2023; Zelikman et al., 2022) to achieve significant boost in capabilities, especially in mathematics problem solving and code generation.

In this work we explore improving planning and reasoning capabilities of a web agent, which interacts with a real world website. Our goal is to design an approach that allows the agent to improve with autonomous experience and limited supervision. Indeed, prior works (Yao et al., 2023b; Zhang et al., 2024c; Masterman et al., 2024; Sumers et al., 2024) have shown strong reasoning to be critical for performance of autonomous agents, where challenges are even greater than during text generation, as the model needs to further understand how its actions affect its environment. Towards this goal, we introduce **Agent Q**—a novel approach that combines several key concepts in reasoning, search, self-critique and reinforcement learning. Our method takes inspiration from Sutton's The Bitter Lesson on the power of general purpose methods that continue to scale with increased computation, showing the significant benefits of combining *search* and *learning*.

Inspired by the success of search-based methods in prior game-playing settings (Silver et al., 2017a; Brown and Sandholm, 2019; Gray et al., 2021) and mathematical reasoning (Yao et al., 2023a; Besta et al., 2024), we deploy a Monte Carlo Tree Search (MCTS) based search routine over web pages to guide agent exploration. Given the complexity of the environment, we use a base LLM for sampling possible rationales and web actions to explore. While this simple search-strategy shows a meaningful improvement in the success rate, it still struggles on long horizon tasks due to sparsity of environment rewards. Indeed even a small mistake across the trajectory can cause the final agent output to be wrong, creating significant credit assignment problems. To overcome this, we use AI feedback (Bai et al., 2022) and self-criticism (Yuan et al., 2024) to further prompt the LLM to provide self-evaluation feedback at each node, which serves as intermediate reward and helps guide the search steps. This meaningfully improves the final agent success rate, but requires significant online interactions and moreover the capability to rollback actions, which is not always possible in online realistic settings. Such online autonomous search with little supervision on the web can result in a weak or unsafe agent which can make many errors, resulting in risky behaviors in sensitive online settings like bank transfers and sensitive information sharing.

To correct this, we use the traces generated by the search process to improve capabilities of the model by learning from both the successful and unsuccessful trajectories with offline reinforcement learning, utilizing the Direct Preference Optimization (DPO) algorithm. We create preferences over different branches at the node level, which are scored using a mixture of the AI process feedback rewards and the final success rate of the explored branch. We evaluate our approach on the simulated WebShop benchmark (Yao et al., 2022)—a simulated e-commerce platform—as well as a real-world reservations booking website. We utilize LLaMa 3-70B as the base model in our experiments. In the WebShop environment, our approach consistently outperforms behavior cloning and reinforcement learning fine-tuned baselines, and **beats average human performance when equipped with the capability to do online search.**

In our real-world booking experiments, using our **Agent Q** framework we improve the model zero-shot absolute success rate from **18.6%** to **81.7%** (a **340% relative increase**), outperforming GPT-4's performance after a single day of autonomous data collection. When we equip Agent Q with online search capability, our absolute success further improves to **95.4%**. We believe that our approach represents a significant step forward in the development of autonomous web agents through it's search and self-critique capabilities, setting a new benchmark for reliable multi-step decision-making in interactive settings.

## 2 RELATED WORK

Our work touches on a large number of research directions around agent design, self-improvement, reasoning and reinforcement learning. We include a short overview of related works from those various fields below.

## 2.1 GUIDED SEARCH FOR REASONING AND PLANNING

The latest generation of Large Language Models (LLMs) have demonstrated promising emerging properties around reasoning and planning. Moreover such behaviours can be directly elicited from strong models only using simple prompting techniques (Wei et al., 2022; Kojima et al., 2022; Qiao et al., 2023). These have also become an integral part of agentic design (Yao et al., 2023b; Zhang et al., 2024c), which we also utilize for our approach. Another emerging research direction is based around step-by-step verifiers or "Process Reward Models" (Uesato et al., 2022; Lightman et al., 2023), specifically for mathematical reasoning. These have shown to improve performance beyond purely outcome-based training, however they require a large amount of human effort to label individual steps. Some recent approaches have proposed self-supervised methods for step-level supervision (Hwang et al., 2024; Wang et al., 2024b; Setlur et al., 2024a). A number of concurrent works (Xie et al., 2024; ?; Tian et al., 2024) have further explored tree-based search approaches (Yao et al., 2023a) in combination with DPO (Rafailov et al., 2023) training for math-based reasoning. These algorithms optimize actions at the node level, using different branches produced by the search algorithm to create preference pairs. Our approach shares similarities to the self-supervised search proposed in (Yao et al., 2023a) with a combination of AI-based feedback (Bai et al., 2022; Yuan et al., 2024) to guide intermediate search steps, but we are the first to scale this a realistic agent setting. Similar approaches were proposed in (Zhou et al., 2024a; Hao et al., 2023; Kang et al., 2024), and other works (Koh et al., 2024); however these works only use the base model's zero-shot capability to search and do not train it further. Moreover they are only evaluated on simulated environments. Beyond the search stage, our work further adopts the training methodology of (Xie et al., 2024; Zhang et al., 2024e; Tian et al., 2024), which significantly boosts our agent's zero-shot capabilities.

## 2.2 WEB AGENTS

The strength and capabilities of recent pretrained Large Language (Vision) Models LL(V)Ms has significantly boosted progress in developing autonomous web-agents. Improved code understanding and long context have allowed agents to represent environment state and action space with document object model (DOM) allowing for deployment in complex and realistic domains. Moreover strong reasoning (Yao et al., 2023b) and planning (Liu et al., 2023; Zhang et al., 2024c) capabilities have also led to the development of a number of promising agents (Zhang and Zhang, 2023; Hong et al., 2023; Zhou et al., 2024b; Deng et al., 2023; Gur et al., 2024). Beyond using LL(V)Ms as plug-and-play planners/policies, recent works have sought to improve agentic-specific performance. Examples include online exploration (Zhang et al., 2024a), planning (Zhang et al., 2024b), error-correction (Wang et al., 2024a), and self- (Wu et al., 2024) or AI-critique (He et al., 2024; Pan et al., 2024). However, with small exceptions (Nakano et al., 2022) (which is still limited in scope) these agents mostly provide a framework around a strong pre-existing model like GPT4-V or deploy limited fine-tuning and adaptation. In this work we show that model training is crucial for continuous improvement. We combine a planning and reasoning agent with MCTS inference-time search and AI self-critique for self-supervised data collection, which we then use for RL type training.

## 2.3 REINFORCEMENT LEARNING FOR LLMS AND AGENTS

Reinforcement Learning has become a significant component of training modern generative AI systems (Ouyang et al., 2022; Bai et al., 2022; Touvron et al., 2023). Classical approaches have deployed the PPO algorithm (Schulman et al., 2017)—or similar policy-gradient based methods— and have even been scaled to autonomous web search agents (Nakano et al., 2022) as well as embodied applications with vision-language models (Zhai et al., 2024) (in simulation). However, these algorithms are challenging due to their complexity and the need for a high number of online samples from the model. This is especially prominent in potentially risky situations, such as autonomous agentic models that could make a number of impactful mistakes during training. Implicit Language Q-learning (Snell et al., 2022) and the Q-transformer (Chebotar et al., 2023) are offline RL algorithms (Levine et al., 2020) designed for auto-regressive transformer models, and hence can be safely trained on pre-collected datasets; however they have not been successfully scaled to modern LLMs. While these methods represent a token-level MDP, (Zhou et al., 2024c) has shown success formulating the RL problem at a step level and these ideas have recently been scaled to a general device-control agent (Bai et al., 2024). However, these algorithms still have high complexity and require auxiliary models, such as value functions, so instead in our approach we opt to use the Direct

Preference Optimization (DPO) algorithm (Rafailov et al., 2023) due to it's simplicity and natural fit for the branching nature of tree-search based data.

# 3 PRELIMINARIES

In this section we will outline the preliminaries of our agent training process. For a full description of our agentic system formulation consider Appendix A. For training purposes at each time step $t$ the agent receives a state $\mathbf{h}_t$ and will produce actions $\mathbf{a}_t \sim \pi(\mathbf{a}|\mathbf{h}_t)$.

## 3.1 FINE-TUNING LANGUAGE MODELS FROM FEEDBACK

Classical approaches to RLHF in foundation models (Stiennon et al., 2022; Ouyang et al., 2022) use the model as a policy $\pi_\theta$ and optimize an objective of the form:

$$\mathbb{E}_{\mathbf{a}\sim\pi_\theta(\mathbf{a}|\mathbf{h})}[r(\mathbf{a}, \mathbf{h})] - \beta\mathbb{D}_{KL}[\pi_\theta(\mathbf{a}|\mathbf{h})||\pi_{\text{ref}}(\mathbf{a}|\mathbf{h})] \tag{1}$$

where $\pi_{\text{ref}}$ is some reference policy (usually the initial model). The goal of this formulation is to optimize some target objective (expressed by the reward $r(\mathbf{a}, \mathbf{h})$) while preventing out-of-distribution drift. This objective can be extended to multi-step agentic problems, where the model interacts with an external environment env such as in Nakano et al. (2021) which focuses on information retrieval using web navigation. In this case we use an objective of the kind

$$\mathbb{E}_{\pi_\theta,\text{env}}\left[\sum_t r(\mathbf{a}_t, \mathbf{h}_t) - \beta\mathbb{D}_{KL}[\pi_\theta(\mathbf{a}_t|\mathbf{h}_t)||\pi_{\text{ref}}(\mathbf{a}_t|\mathbf{h}_t)]\right] \tag{2}$$

Classical RLHF has used policy gradient type of algorithms, such as PPO (Schulman et al., 2017), however, they are complex and require online data, which can be costly/dangerous to collect autonomously in the agent setting. While PPO has shown some success in prior web agent applications (Nakano et al., 2021). The issues above largely make the approach not practical for general web tasks, beyond information retrieval. In this work we utilize some recent alternatives, outlined below.

### 3.1.1 REINFORCED FINE-TUNING

Reinforced fine-tuning (RFT) algorithms (Zelikman et al., 2022; Gulcehre et al., 2023; Yuan et al., 2023; Singh et al., 2024) have grown in popularity due to their simplicity and scalability. These methods aggregate data and filter out the sub-optimal samples based on some reward model or a verifier to construct a growing dataset of high-quality trajectories $\mathcal{D}$. Given this dataset and a parameterized model $\pi_\theta$ we can carry out standard supervised fine-tuning (SFT):

$$\mathcal{L}(\pi_\theta, \mathcal{D}) = -\mathbb{E}_\mathcal{D}\left[\sum_{t=1}^T \log \pi_\theta(\mathbf{a}_t|\mathbf{h}_t)\right] \tag{3}$$

In this objective the divergence penalty is only applied implicitly by limiting the number of training rounds. While simple and relatively successful, empirically these methods tend to under-perform standard RL and alternatives (Dubois et al., 2024; Tajwar et al., 2024; Setlur et al., 2024b) in the text generation domain, particularly in reasoning. We largely observe similar empirical results, and we use these methods mostly as baselines to build intuition.

### 3.1.2 DIRECT PREFERENCE OPTIMIZATION

Direct Preference Optimization (DPO) (Rafailov et al., 2023) is an offline RL (Levine et al., 2020) alternative to the classical RLHF optimization pipeline. It is a suitable algorithm for agent fine-tuning, as it can use fully offline data and does not require online rollouts. The original formulation in the pure text generation setting is based on the RL problem in Eq. 1 and considers feedback of pairwise comparisons $(\mathbf{h}, \mathbf{a}^w, \mathbf{a}^l)$, where $\mathbf{h}$ is a single prompt and $\mathbf{a}^w$ and $\mathbf{a}^l$ are two responses with $\mathbf{a}^w \succ \mathbf{a}^l$ indicating that $\mathbf{a}^w$ is preferred over $\mathbf{a}^l$. The DPO objective then minimizes the following loss:

$$\mathcal{L}_{\text{DPO}}(\pi_\theta; \mathcal{D}) = -\mathbb{E}_{(\mathbf{h},\mathbf{a}^w,\mathbf{a}^l)\sim\mathcal{D}}\left[\log \sigma\left(\left(\beta \log \frac{\pi_\theta(\mathbf{a}^w|\mathbf{h}^w)}{\pi_{\text{ref}}(\mathbf{a}^w|\mathbf{h}^w)}\right) - \left(\beta \log \frac{\pi_\theta(\mathbf{a}^l|\mathbf{h}^l)}{\pi_{\text{ref}}(\mathbf{a}^l|\mathbf{h}^l)}\right)\right)\right] \tag{4}$$

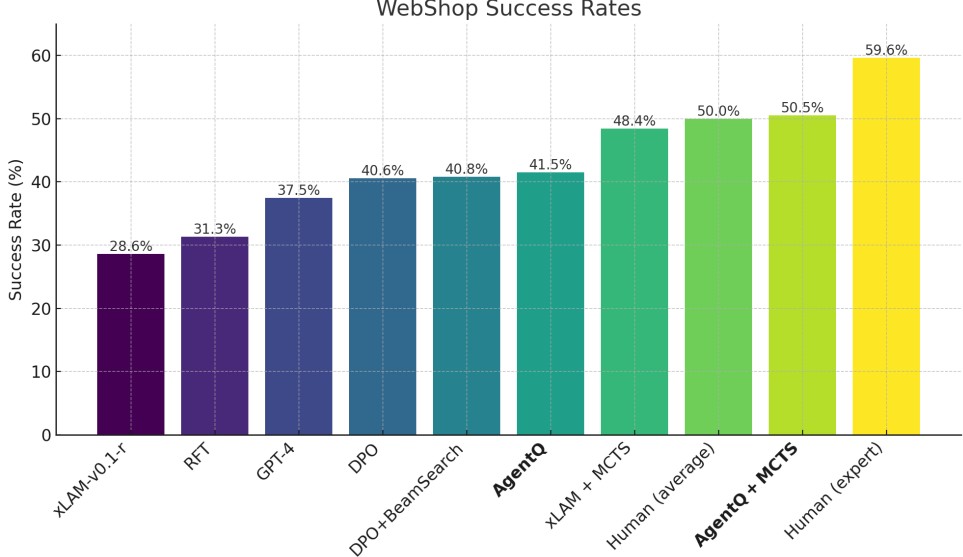

Figure 1: Success rate of different approaches on the WebShop tasks (Yao et al., 2022). All models are based on xLAM-v0.1-r (Zhang et al., 2024c). RFT and DPO over xLAM-v0.1-r demonstrate improvements in performance from 28.6% to 31.3% and 40.6% respectively. However, these methods still lag behind average human performance of 50.0%. Our approach, Agent Q + MCTS achieves a significant gain (76.57% relative improvement) over the base model, outperforming average human performance on WebShop with a success rate of **50.5%**.

While the algorithm was developed in a bandit setting, Hejna et al. (2024); Rafailov et al. (2024) have extended it to the multi-turn setting in Eq. 2 with preferences over trajectories. In our setting, we can directly utilize this objective as:

$$\mathcal{L}_{\text{T-DPO}}(\pi_\theta; \mathcal{D}) = -\mathbb{E}_{(\tau^w, \tau^l) \sim \mathcal{D}} \left[ \log \sigma \left( \left( \sum_{t=0}^{|\tau^w|} \beta \log \frac{\pi_\theta(\mathbf{a}_t^w | \mathbf{h}_t^w)}{\pi_{\text{ref}}(\mathbf{a}_t^w | \mathbf{h}_t^w)} \right) - \left( \sum_{t=0}^{|\tau^l|} \beta \log \frac{\pi_\theta(\mathbf{a}_t^l | \mathbf{h}_t^l)}{\pi_{\text{ref}}(\mathbf{a}_t^l | \mathbf{h}_t^l)} \right) \right) \right]$$
(5)

One bottleneck for the practical deployment of the algorithm is the need for a reference model $\pi_{\text{ref}}$ during optimization, which requires more computational resources. Instead, in our settings, we slightly modify the algorithm using an off-policy replay buffer, which aggregates trajectory data, as well as likelihoods of the generated actions. During the optimization step, we sample trajectory pairs $(\tau^w, \tau^l)$ where $\tau^w \succ \tau^l$, as well as the corresponding likelihoods under the data generation (reference) density. This eliminates the need for a separate reference model while training.

## 4 PRELIMINARY APPROACH WITH OUTCOME SUPERVISION

In this section we will outline preliminary experimental results, which will build the base understanding for our further experiments. We use the AgentOhana xLAM-v0.1-r model (Zhang et al., 2024c), which is a fine-tune of a pre-trained Mixtral-8x7B-Instruct-v0.1 model (Jiang et al., 2024) on a mix of agentic applications, including WebShop SFT data. We also incorporate the same agent configuration[1] specified by AgentLite (Liu et al., 2024) to ensure a fair comparison between our fine-tuned model and the xLAM base model performance. We evaluate all approaches on the WebShop environment (Yao et al., 2022), where the agent needs to find particular products by browsing a simulated web shop. The environment comes with a set of 12,087 pre-defined tasks (corresponding to specific products to find), which we split into a train set of 11,000 tasks, which we use for further agent fine-tuning and a set of 1,087 held-out tasks, which we use for zero-shot evaluation. We show success rates (exact product match) for different approaches in Figure 1. The base xLAM-v0.1-r model achieves success rate of 28.6% on the test tasks. All other methods are based on outcome-based supervision

---

[1]https://github.com/SalesforceAIResearch/xLAM

only, depending on whether a particular attempt was successful or not. We see that further RFT training, using a STaR-like algorithm (Zelikman et al., 2022) on the trajectory level, as outlined in Sec. 3.1.1, achieves success rate of 31.3%, which is a small improvements of 2.7% over the initial model. This is not surprising since the base model is already trained as an agent on the environment with supervised fine-tuning on demonstrations. Our next experiment fine-tunes the base model using the trajectory-level DPO algorithm, as outlined in Eq. 5 in Sec. 3.1.2 using successful trajectories as preferred over failed ones. This approach also uses only outcome-level supervision, but unlike the RFT baseline can utilize failed trajectories as well, which improves the agent performance by 9.3% over RFT agent to 40.6% success rate. We also evaluate this model with beam search for the action generation, which can be considered a form of planning the horizon of a single environment action (which still consists of multiple simple actions) (Rafailov et al., 2023), but it only yields marginal improvement over the base model. These findings match results on reasoning for math problems (Pang et al., 2024) and some recent approaches that also apply DPO to agent applications (Song et al., 2024; Xi et al., 2024).

Despite the additional reinforcement learning training, these agents are still not able to match the average human performance on this environment. We identify that one of the core failure modes of the DPO policy is that it executes a greedy search when looking for matches to the product query. For example, for every search query, the WebShop environment yields a number of pages of results. However, we find that the model nearly always greedily searches for the best matching item in the first page of results rather than using the "[NEXT]" and "[PREV]" buttons to navigate between pages, essentially deploying a weak exploration strategy.

## 5 AGENT SEARCH

As we discovered in the previous section, while training based on outcome supervision with DPO yields meaningful improvement, the model is still not able to match human performance due to its limited exploration. In this section we introduce AgentQ, which endows agent with additional search and learning capabilities. The base AgentQ model uses MCTS Guided Direct Preference Optimization to learn how to perform web agent tasks at the step-level. We also introduce AgentQ+MCTS, which additionally uses inference time MCTS algorithm to further improve performance.

### 5.1 MONTE-CARLO TREE SEARCH OVER WEB-PAGES

The Monte Carlo Tree Search (MCTS) algorithm (Kocsis and Szepesvári, 2006) employed in this work follows closely the one in Hao et al. (2023) and consists of four phases: selection, expansion, simulation, and backpropagation. Each phase plays a critical role in balancing exploration and exploitation while iteratively refining the policy.

We formulate the web agent execution as tree search over web-pages. The state is represented as described in Appendix A and consist of the summary of the agent's history and the DOM tree of the current web-page. Unlike board games, such as Chess or Go (Silver et al., 2017b) the complex web-agent action space we use is open-format and variable. Instead we will use the base model as an action-proposal distribution and sample a fixed amount of possible actions at each node (web-page). Once we select and execute an action in the browser we traverse the next web-page, which together with the updated history becomes the new node.

### 5.1.1 ACTION SELECTION WITH AI PROCESS SUPERVISION

The selection phase uses the Upper Confidence Bound (UCB1) formulation of MCTS, also used by Hao et al. (2023), to select nodes with the aim to balance exploration and exploitation. With some abuse of notation we will also denote the agent state with $\mathbf{h}_t$. We consider the value function $Q(\mathbf{h}_t, \mathbf{a})$ which represents the estimated value (chance of success) of taking action $\mathbf{a}$ in the state $\mathbf{h}_t$. At each new node $\mathbf{h}_t$ we sample $K$ proposal actions from the base model $\mathbf{a}_t^1, \ldots, \mathbf{a}_t^K$. We initialize all values $Q(\mathbf{h}_t, \mathbf{a}_t^i), i = 1, \ldots, K$ to zero. The web-based environment does not provide intermediate rewards to guide the search, so we incorporate AI-based critique to provide process supervision at the step level to guide the exploration process. We use the base model to produce a feedback score for each action by asking it to rank the generated actions by its perceived utility in helping the agent complete the user task.

Figure 2: The policy proposes $K$ actions at every step during inference time search. The critic, also initialized as the same base LLM model used by the policy, ranks the actions proposed by the policy. This ranking is used to guide node selection after expansion and used to construct preference pairs during policy training.

We query the feedback model for multiple iterations, each time removing the best action selected from the previous iteration from the list, until we have a full ranking of all actions. The full AI feedback process is demonstrated in Figure 2. The AI feedback is used for the initial ordering of actions to explore as well as later for collecting the preference pairs. After the initial selection, we select actions to explore based on the standard MCTS UCB1 formulation:

$$\mathbf{a}_t^* = \arg \max_{\mathbf{a}_t^1, \ldots, \mathbf{a}_t^K} \left[ Q(\mathbf{h}_t, \mathbf{a}) + c_{\exp} \cdot \sqrt{\frac{\log N(\mathbf{h}_t)}{1 + N(\mathbf{h}_{t+1})}} \right], \quad (6)$$

where $N(\mathbf{h}_t)$ is the visitation frequency of state $\mathbf{h}_t$, and $c_{\exp}$ is an exploration constant. For each rollout added to the tree, we start at the root node and follow the child states that maximize the UCB1 score until we reach a leaf node. This process is repeated for each tree/prompt in the batch.

### 5.1.2 EXPANSION AND BACKTRACKING

Based on the preceding section, we select and execute an action in the browser environment to reach a new node (page). Beginning from the selected state node's trace, we roll out the trajectory using the current policy $\pi_\theta$ until a terminal state is reached. The environment returns a reward at the end of the trajectory, $R$, where $R = 1$ if the agent was successful and $R = 0$ otherwise. We then backpropagate this reward by updating the values of each node bottom up from the leaf node to the root as follows:

$$Q(\mathbf{h}_t, \mathbf{a}_t^i) \leftarrow \frac{Q(\mathbf{h}_t, \mathbf{a}_t^i)N(\mathbf{h}_t, \mathbf{a}_t^i) + R}{N(\mathbf{h}_t, \mathbf{a}_t^i) + 1}$$

$$N(\mathbf{h}_t, \mathbf{a}_t^i) \leftarrow N(\mathbf{h}_t, \mathbf{a}_t^i) + 1 \quad (7)$$

Each state node tracks two values: $Q(\mathbf{h}_t, \mathbf{a}_t^i)$, the average reward for passing through state $\mathbf{h}_t$ and choosing action $\mathbf{a}_t^i$, and $N(\mathbf{h}_t, \mathbf{a}_t^i)$, the number of times this state action pair was visited during search (and $N(\mathbf{h}_t) = \sum_{i=1}^{K} N(\mathbf{h}_t, \mathbf{a}_t^i)$). The backpropagation updates correctly maintain these values.

### 5.2 IMPROVING ZERO-SHOT PERFORMANCE WITH REINFORCEMENT LEARNING

Training large foundation models with offline (Snell et al., 2022) or off-policy (Chebotar et al., 2023) reinforcement learning at scale has still remained challenging. At the same time online (on-policy) reinforcement learning (Stiennon et al., 2022; Ouyang et al., 2022) is not scalable to real interactive environments. Instead, we follow a line of recent works, which apply the DPO algorithm (Rafailov et al., 2023; 2024) at the step level in multi-step reasoning problems in mathematical domains (Xie et al., 2024; Hwang et al., 2024; Chen et al., 2024; Lai et al., 2024b; Lu et al., 2024; Setlur et al., 2024b; Zhang et al., 2024e). Our approach is most similar to Xie et al. (2024); Chen et al. (2024); Zhang et al. (2024e) who also use the branching nature of tree search to produce step-level preference pairs. We will also use this approach in our setting due to its simplicity, scalability and prior success in smaller scale (non-interactive) reasoning applications.

---

**Algorithm 1** MCTS Guided Direct Preference Optimization

---

**Input:** $\pi_{\theta_0}$: initial LLM policy, $\mathcal{D}_T$: dataset of tasks the agent must complete in the environment, $N$: number of iterations, $B$: number of samples per iteration, $T$: MCTS tree depth, $\mathcal{B}$: replay buffer, $\theta_{\text{threshold}}$: value threshold in Eq. 9, $K$: number of actions to sample for MCTS

**Output:** $\pi_{\theta_N}$, the trained LLM policy

**for** $i = 1$ to $N$ **do**

    $\pi_{\text{ref}} \leftarrow \pi_{\theta_i}, \pi_{\theta_i} \leftarrow \pi_{\theta_{i-1}}$

    Sample a batch of $B$ tasks from $\mathcal{D}_T$

    **for** each task in batch **do**

        Initialize the root node $\mathbf{h}_0$

        **for** $t = 1$ to $T$ **do**

            **Selection:** Traverse tree from the root node to a leaf node using tree policy (UCB1; 6)

            **Expansion:** From the selected node, sample $K$ actions using $\pi_{\theta_i}$ and rank with

                AI Process Supervision (Sec 5.1.1)

            **Trajectory Rollout**: From the selected node's trace, roll out the trajectory using

                $\pi_{\theta_i}$ until a terminal state is reached

            **Backpropagation:** Backpropagate the value estimate bottom-up (Eq. 7)

        **end for**

        Collect trajectories from rollouts and store them in replay buffer $\mathcal{B}$

    **end for**

    Construct preference pairs $\mathcal{D}_P = \{(\mathbf{h}_t, \mathbf{a}_t^w, \mathbf{a}_t^l)\}_{t=1}^{T-1}$ where $\mathbf{h}_t \sim \mathcal{D}_P$. For each node at step level $t$, compare each pair of child nodes, and construct the pair of generated actions $(\mathbf{a}^w, \mathbf{a}^l)$ if the values of taking the action, $|Q(\mathbf{h}_t, \mathbf{a}^w) - Q(\mathbf{h}_t, \mathbf{a}^l)| > \theta_{\text{threshold}}$, where $Q(\mathbf{h}_t, \mathbf{a}^w)$ and $Q(\mathbf{h}_t, \mathbf{a}^l)$ are computed using Eq. 9

    Optimize LLM policy $\pi_{\theta_i}$ using DPO objective in Eq. 4 with $\mathcal{D}_P$ and $\pi_{\text{ref}}$

**end for**

---

We will generate a dataset of preference pairs $\mathcal{P} = \{\mathbf{h}_t, \mathbf{a}_t^w, \mathbf{a}_t^l\}$ where we make sure both actions were explored. We then optimize the DPO objective in Eq. 4 on the node level. We will leverage a theoretical result below to guide the construction of these preferences. We can make a number of modifications to Theorem 6.1 from Setlur et al. (2024b) to incorporate the interactive nature of the web environment dynamics to obtain the following result:

**Theorem 1.** *Consider a policy that optimizes the objective in Eq. 2 on trajectories generated by $\pi_{ref}$ and that at each node $\mathbf{h}_t$ we have preferences generated accordingly to $p(\mathbf{a}_t^w \succ \mathbf{a}_t^l | \mathbf{h}_t) \propto \sigma(Q(\mathbf{h}_t, \mathbf{a}_t^w) - Q(\mathbf{h}_t, \mathbf{a}_t^l))$, then the policy which optimizes the DPO objective in Eq. 4 is identical to the optimal RL policy*

$$\pi^*(\mathbf{a}|\mathbf{h}_t) \propto \pi_{ref}(\mathbf{a}|\mathbf{h}_t) \exp\left(Q(\mathbf{h}_t, \mathbf{a})/\beta\right) \tag{8}$$

*Proof.* The proof follows directly from the proof of Theorem 6.1 in Setlur et al. (2024b) and the control as inference arguments in Rafailov et al. (2024); Levine (2018). □

That is, we can approximate the optimal RL policy if we generate preferences under the optimal value function (or an approximation thereof). Since the outcome success provides limited supervision we also incorporate process supervision through the AI feedback as outlined in Section 5.1.1. We interpret the ranking of possible actions by the model to be driven by an implicit value function. Similar semantics was used in Koh et al. (2024), where GPT-4 was used as a zero-shot value function, while here we ask the model to instead reason over the given potential actions and provide rankings instead. This self-rewarding approach has shown promise in the RLHF setting (Yuan et al., 2024) and we utilize it for our agent setting as well. Under this formulation, we compute the state-action value as an average:

$$Q(\mathbf{h}_t, \mathbf{a}_t^i) = \alpha \tilde{Q}(\mathbf{h}_t, \mathbf{a}_t^i) + (1 - \alpha)\hat{Q}(\mathbf{h}_t, \mathbf{a}_t^i) \tag{9}$$

where $\tilde{Q}(\mathbf{h}_t, \mathbf{a}_t^i)$ is the empirical value estimated through MCTS backpropagation and $\hat{Q}(\mathbf{h}_t, \mathbf{a}_t^i)$ is a value estimate based on the ranking of the action $\mathbf{a}_t^i$ by the process supervision AI model. Specifically, we treat the lowest ranked action as having a $\hat{Q}(\mathbf{h}_t, \mathbf{a}_t^i)$ estimate of 0.0 and the highest ranked action as having a $\hat{Q}(\mathbf{h}_t, \mathbf{a}_t^i)$ estimate of 1.0, and interpolate the actions in between based on their ranking.

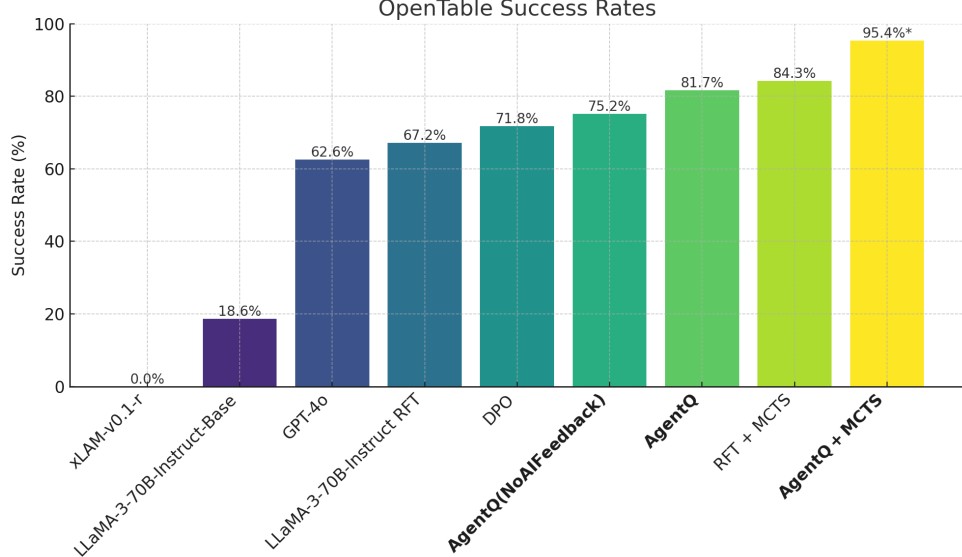

Figure 3: Success rate of different approaches on OpenTable. All models unless otherwise stated are based on LLaMA-3-70B-Instruct (Touvron et al., 2023). Using DPO and RFT with MCTS further improves performance from 18.6% to 71.8% and 84.3% respectively. We show that Agent Q in itself achieves 81.7% and Agent Q + MCTS significantly outperforms all other techniques, with a performance of **95.4%** on OpenTable.

Finally, we create the preference dataset over pairs of actions for which the difference in value, $|Q(\mathbf{h}_t, \mathbf{a}_t^w) - Q(\mathbf{h}_t, \mathbf{a}_t^l)|$, is greater than the value threshold hyperparameter, $\theta_{\text{threshold}}$. The full outline of our RL approach is shown in Algorithm 1.

## 6 RESULTS

### 6.1 FULL WEBSHOP RESULTS

The full range of results and baselines is shown in Figure 1. The headline result is that Agent Q with test-time MCTS search (Agent Q + MCTS) is able to slightly outperform the average human success rate. When just looking at agents that do not use test-time search, we see that training approach outlined in Algorithm 1, gives Agent Q an improvement of 10.2% over RFT and a 0.9% improvement over DPO from outcome supervision. We note that while our dense-level supervision improves over purely outcome-based one, the improvement is modest on WebShop. This is because the WebShop environment requires relatively short trajectories, and the model is capable enough to learn credit assignment purely from outcome supervision. We will further explore more complex real world environment, which requires longer-range credit assignment. Beyond the zero-shot agents, we see that the ability to search at test time is a significant paradigm shift. Using MCTS on top of the base xLAM-v0.1-r model, significantly boosts success rates from 28.6% to 48.4%, approaching close to the average human performance of 50.0% and significantly out-performing the zero-shot performance of the DPO model. As mentioned before, pairing test-time MCTS with the trained Agent Q model improves performance to 50.5%, slightly beating the average human success rates.

### 6.2 SCALING TO REAL WORLD WEBSITES

In this section we will investigate scaling the Agent Q framework to real use cases on live websites, in particular bookings on OpenTable. Initial experiments showed that the xLAM-v0.1-r model was too weak for the task, achieving a success rate of 0.0%. Instead, we use LLaMa3-70B-Instruct, which achieved non-trivial success rates. Descriptions of our real-world environment are in Appendix B.

The base xLAM-v0.1-r model achieves a success rate of 0.0%, largely from failing to follow instructions for the general web navigation instructions used for live websites, contrary to the simplified

observation and action space used in WebShop. We instead initialize the base policy with the LLaMa-3 70B Instruct model, which achieves a zero-shot success rate of 18.6%. We do a single round of RFT on 600 successful trajectories which improves the success rate to 67.2% already out-performing the the GPT-4o model zero-shot performance with a success rate of 62.6%. For all other baselines we adopt the RFT model as the reference policy, due to the relatively low success rate of original LLaMa 3 70B Instruct model.

In this environment, training with outcome-supervision only DPO further improves performance by 4.6% to 71.8% but significantly under-performs the full Agent Q pipeline which achieves a zero-shot success rate of 81.7% We hypothesizes that this is due to the fact that OpenTable is a significantly more challenging environment, which requires almost twice as many steps to complete as WebShop, so the agent benefits from fine-grained supervision and credit assignment. We further ablate the role of the intermediate AI feedback process supervision during training as outlined in Eq. 9 and use MCTS with online Q values computed from outcome rewards only. This setting still outperforms training with trajectory-level DPO (75.2% versus 71.8%) likely due to the more fine-grained credit assignment that the branching tree search provides to the agent. However, zero-shot performance is still meaningfully worse than using intermediate process-level supervision and the full Agent Q achieves 6.5% higher success rate at 81.7%.

Similar to the WebShop experiment we see a step level increase in capability from allowing the model to search at inference time, with the base RFT model achieving 84.3% success with MCTS, outperforming the Agent Q zero-shot performance of 81.7% success. However, if we carry out additional MCTS search using the Agent Q model as the base policy we achieve a significant 95.4% success rate.

## 7 DISCUSSION

In this work we developed algorithms for autonomous improvement of web-agents with limited human supervision. While most prior works build frameworks around existing models without additional training, we specifically seek to fine-tune pre-trained models for web navigation tasks based on synthetic reasoning and search data. While we achieve significant improvement in model capabilities on our target domain, many research questions remain.

**Design of reasoning algorithms.** The core challenge for our web agents is the weak reasoning capabilities, which limit the agent's exploration and search strategy. In our approach we used process-level supervision from a separate critic model, which we prompt to rank possible agent actions. This is in contrast to works in mathematical reasoning where PRMs are usually trained to classify the correctness of individual steps (Lightman et al., 2023), while other agent works (Koh et al., 2024) have prompted models as zero-shot value functions. Furthermore, while we spent significant effort in training the agent policy, we maintain a frozen critic, which would likely also benefit from additional fine-tuning. We defer exploration of these design choices to further work.

**Choice of search algorithm.** We used MCTS search due to the approach's prior success in mathematical and code reasoning tasks. However, agent models executing MCTS on live environments might require significant number of risky interactions and a different search strategy might be more suitable. Recent works such as Lehnert et al. (2024); Gandhi et al. (2024) have even suggested directly learning to optimally search and explore in reasoning tasks using meta-reinforcement learning. We believe this is a promising research direction for autonomous agents, which we will pursue in further work.

**Discrepancy between zero-shot vs search results.** Similar to some recent works that focus on code and reasoning, we observe significant gap between zero-shot agent performance and performance of the agent equipped with search capabilities (Snell et al., 2024; Brown et al., 2024). Investigating these trade-offs at scale and the potential effect of different search/optimization approaches.

**Online safety and interaction.** The design of agent Q allows for largely autonomous exploration, self-evaluation and improvement with limited human intervention. However, the agent might make a significant number of mistakes in it's search process which might be difficult to fix/reverse, especially for safety-critical online transactions, such as communications/email, payments, filings etc. This limits the scope of websites that Agent Q can be safely deployed and we might require additional safety critics and human-in-the-loop training setups.

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

**Agent Input**

```
<SYSTEM PROMPT>
<EXECUTION HISTORY>

USER QUERY: Book a reservation for the
restaurant Ceconni's on OpenTable for
2 people on June 17 2024 at 7:00pm

CURRENT OBSERVATION:
```

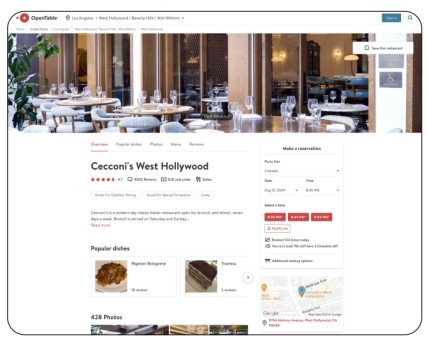

**Agent Output**

```
PLAN:
1. Select the date (May 22, 2024) for the
   reservation.
2. Choose the time (7:00 PM) for the
   reservation.
3. Select the party size (4 people) for the
   reservation.
4. Click on the "Find a table" button to book
   the reservation.

THOUGHT: I am currently on the Cecconi's
restaurant page on OpenTable, and I need to
select the date and time for the reservation
and choose the number of guests. I will focus
on selecting the date, time, and party size
for the reservation.

COMMANDS: CLICK <select>Date</select>

STATUS: CONTINUE
```

Figure 4: We provide the following input format to the Agent, consisting of the system prompt, execution history, the current observation as a DOM representation, and the user query containing the goal. We divide our Agent output format into an overall step-by-step plan, thought, a command, and a status code.

## A  AGENT FORMULATION

We consider a general POMDP setup $(\mathcal{O}, \mathcal{S}, \mathcal{A}, T, R, \mu_0, \gamma)$ where $\mathcal{O}$ denotes the observation space, $\mathcal{S}$ the unobserved state space, $\mathcal{A}$ the action space, $T(\mathbf{s}_{t+1}|\mathbf{s}_t, \mathbf{a}_t)$ the transition distribution (in this case the dynamics of a web browser), $R(\mathbf{s}, \mathbf{a})$ the reward function (in this work we use sparse rewards of 1/0 representing success/failure), $\mu_0(\mathbf{s}_0)$ the initial state distribution, and $\gamma$ the discount factor, which we set to 1. A POMDP is the most suitable framework to model web interactions for several reasons - first novel environments, which the agent is unfamiliar with require exploration in order to locate the task objective, consistent with the meta-reinforcement learning as task inference view (Humplik et al., 2019). Moreover, the real web is dynamic, which creates partial observability of the current state each time the agent is deployed - i.e. it does not a priori know current booking availability before attempting to do it. We will outline the main parts of our web agent below.

**The agent observation** $\mathbf{o}_t \in \mathcal{O}$ are commands/information given by the user and the web browser. The first observation $\mathbf{o}_1$ is a user text instruction, such as

"Book reservation for restaurant Cecconi's on OpenTable for 4 people on May 22 2024 at 7:00 PM"

for example and a browser home page. Subsequent observations consist of web pages from the browser, represented as a HTML DOM format. Occasionally for some tasks the agent might ask for confirmation/feedback from the user, which then also becomes part of the observation.

**The agent actions** $\mathbf{a}_t \in \mathcal{A}$ are composite, based on agent history $\mathbf{h}_t$. Our base approach is a ReAct agent Yao et al. (2023b) with a preliminary planning step (PlanReAct) Liu et al. (2023) with few additional components.

- **Planning** For the first action after the initial observation we leverage the base LLM's planning capabilities (Huang et al., 2022a) and prompt the agent to generate a plan $\mathbf{a}_1^{\text{plan}} \sim \pi(\mathbf{a}_1^{\text{plan}}|\mathbf{h}_1)$ of sequential steps to execute in language.

- **Reasoning** Subsequently all actions consist of a thought action $\mathbf{a}_t^{\text{tht}} \sim \pi(\mathbf{a}_t^{\text{tht}}|\mathbf{h}_t)$, which is a chain-of-thought reasoning step (Wei et al., 2022).

- **Environment action** Next we generate the browser interaction command $\mathbf{a}_t^{\text{env}} \sim \pi(\mathbf{a}_t^{\text{env}}|\mathbf{h}_t, \mathbf{a}_t^{\text{tht}})$, which consists of a finite set of options like "CLICK [ELEMENT ID]", "SCROLL", "TYPE [CONTENT]" or "ASK USER [CONTENT]" etc.. This is the only part of the action generation, which interacts with the environment.

- **Explanation action** After the environment interaction action has been generated, we additional prompt the model for an explanation action $\mathbf{a}_t^{\text{expl}} \sim \pi(\mathbf{a}_t^{\text{expl}}|\mathbf{h}_t, \mathbf{a}_t^{\text{tht}}, \mathbf{a}_t^{\text{env}})$.

We denote the step action $\mathbf{a}_t$ as a tuple of plan, thought, environment and explanation actions for the first step and thought, environment and explanation actions for subsequent steps. When optimizing models we consider the joint likelihood

$$\log \pi(\mathbf{a}_1|\mathbf{h}_1) = \log \pi(\mathbf{a}_1^{\text{expl}}|\mathbf{h}_1, \mathbf{a}_1^{\text{env}}, \mathbf{a}_1^{\text{tht}}, \mathbf{a}_1^{\text{plan}}) + \log \pi(\mathbf{a}_1^{\text{env}}|\mathbf{h}_1, \mathbf{a}_1^{\text{tht}}, \mathbf{a}_1^{\text{plan}}) +$$
$$\log \pi(\mathbf{a}_1^{\text{tht}}|\mathbf{h}_1, \mathbf{a}_1^{\text{plan}}) + \log \pi(\mathbf{a}_1^{\text{plan}}|\mathbf{h}_1) \tag{10}$$

for the initial action and

$$\log \pi(\mathbf{a}_t|\mathbf{h}_t) = \log \pi(\mathbf{a}_t^{\text{expl}}|\mathbf{h}_t, \mathbf{a}_t^{\text{env}}, \mathbf{a}_t^{\text{tht}}) + \log \pi(\mathbf{a}_t^{\text{env}}|\mathbf{h}_t, \mathbf{a}_t^{\text{tht}}) + \log \pi(\mathbf{a}_t^{\text{tht}}|\mathbf{h}_t)$$

for subsequent actions, unlike some prior works (Zhai et al., 2024), which down-weight the reasoning likelihood.

**The agent state** is the current state of the web, which may mot be observable. In this POMDP formulation we also need to build an agent memory component $\mathbf{h}_t$. Prior works have used the entire trajectory of observations and actions, however HTML DOMs can be hundred of thousands of tokens long. Moreover realistic web-tasks can require many more interactions than static benchmarks such as WebShop (Yao et al., 2022) and WebArena (Zhou et al., 2024b), which most prior works use. This makes it impractical to use full web trajectories due to limited context windows, potential out-of-distribution issues and practical inference speed and cost. Instead, we build the history representation of the agent as $\mathbf{h}_t = (\mathbf{a}_1, \ldots, \mathbf{a}_{t-1}, \mathbf{o}_t)$. That is, the agent history consists of the actions generated so far and the current browser state. With some abuse of notation we will also refer to this as the agent state. Even though only the environment action is used for interacting with the browser, we construct the agent thought and explanation actions to act as a form of inner monologue (Huang et al., 2022b) and adequately represent its state and intentions. This allows us to use a significantly more compact history representation. We should note that, while only the environment action affects the browser state, the planning, reasoning and explanation components affect subsequent decisions due to conditioning. Because of this reason, when we optimize the agent, we compute likelihoods over the composite action.

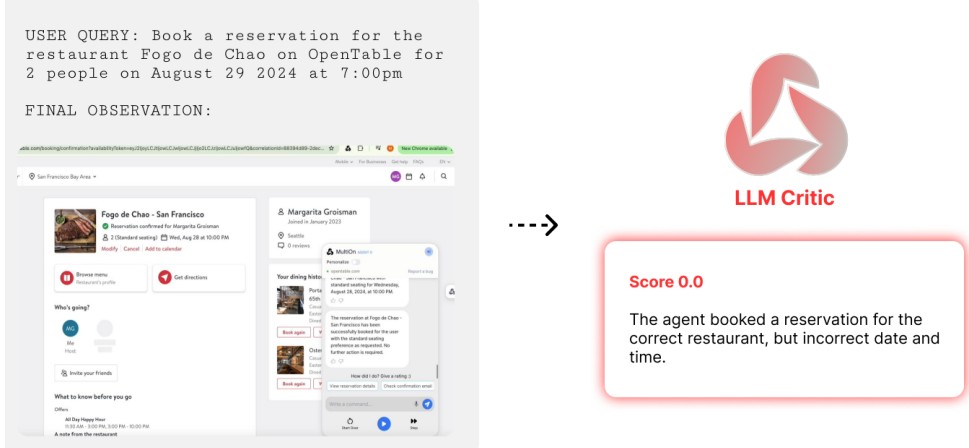

Figure 5: At the end of a trajectory, a GPT-4-V evaluator is called to provide feedback on the agent's performance given the final observation and action history to determine the success score. The model is prompted with a condensed execution history of the trajectory and the screenshot of the final state. The success metric is a binary 0/1 value.

## B OPENTABLE ENVIRONMENT

In OpenTable, the agent is tasked with booking a restaurant reservation for a user. The agent must find a restaurant page on the OpenTable site, look for a reservation at a certain date and time, choose seating options that align with a user's preference and submit the user contact information to complete the task successfully. Since OpenTable is a live environment and is difficult to programatically measure metrics for, we use a language model, GPT-4-V to collect rewards for each trajectory, based on the following metrics: (1) date and time set correctly, (2) party size set correctly, (3) user information entered correctly, and (4) clicked complete reservation. The task is marked as completed if each of the above constraints are satisfied. The outcome supervision setup is shown in Figure 5. We experimented with using LLaMa 70B for outcome supervision as well, but discovered that vision capabilities significantly improve the success classification accuracy (as measured by human validation). At the time of writing no open source vision-language model of sufficient capability was available, hence we opted to use GPT-4-V. We believe that as more open-source multi-modal models become available we can switch to a fully self-supervised pipeline.

To generate queries for the OpenTable benchmark dataset, we programatically generate a diverse set of user queries by combining the restaurant name, desired date and time, and user information.

Navigating on live websites pose a wide variety of challenges. For example, consider that the user specifies a restaurant in a different city than the location the browser is initialized in, the model will have to take extra steps to find the restaurant. Further, if the exact user requested date and time are not available, the model may have to choose the closest available reservation slot. Lastly, if there are preferences, such as indoor or outdoor seating options that the model is presented with, the desired behavior is to interact with the user to determine the best course of action. OpenTable presents a complex set of challenges for web navigation agents; the number of steps required to complete the task is on average 13.9 steps, over double the average number of steps for Webshop, 6.8.

For the observation space for this environment, we design an intermediate state representation that crawls the raw HTML content of a website to retrieve relevant visual components, and highlight interactive elements to the model. The agent is allowed the actions, "CLICK [ID]", "GOTO [URL]", "TYPE [ID] [TEXT]", "SUBMIT [ID]", "CLEAR [ID]", "SCROLL [UP/DOWN]", and "ASK USER HELP". For OpenTable experiments, we use the LLaMA-3-70B-Instruct model as the initial policy. We find that the superior reasoning abilities of this class of model is required for effective task completion, which is necessary to produce the diverse success and failure trajectories required to effectively improve the policy.

