# OpenReview forum: "Agent Q: Advanced Reasoning and Learning for Autonomous AI Agents"
_ICLR.cc/2025/Conference — Submitted to ICLR 2025_

### Official Review · Reviewer_LMY8 · 2024-10-26

**Soundness:** 3
**Presentation:** 4
**Contribution:** 3
**Rating:** 8
**Confidence:** 4

**Summary:**

The paper presents Agent Q, a framework that improves the multi-step reasoning abilities of AI agents in interactive environments by combining guided MCTS with self-critique and an off-policy variant of DPO. It achieves superior performance in web navigation tasks, significantly outperforming standard methods and nearing human-level competence.

**Strengths:**

**Originality:**

The paper introduces a unique approach by combining MCTS with self-critique and DPO to enhance the decision-making abilities of AI agents. This innovative blend of techniques addresses the limitations of traditional LLMs in dynamic environments, making it a fresh contribution to the field.

**Quality:**

The research is thorough and well-executed, with rigorous experiments conducted in both simulated and real-world settings. The significant improvements in success rates, particularly in complex web navigation tasks, demonstrate the robustness and effectiveness of the proposed framework.

**Clarity:**

The paper is clearly written and well-structured, making complex concepts accessible. The methodology and results are presented in a straightforward manner, supported by clear figures and discussions that highlight the framework's advantages.

**Significance:**

Agent Q's ability to outperform existing methods and approach human-level performance in certain tasks is highly significant. It represents a substantial step forward in creating more autonomous and reliable AI systems, with broad implications for various applications, from web navigation to real-world decision-making.

**Weaknesses:**

**Concerns Regarding Multi-turn DPO Loss:**

This paper extends the DPO loss to accommodate multi-turn interactions, as shown in Eq 5. However, recent literature [1], specifically in section 2.2, suggests that the expected multi-turn DPO loss is predicated on deterministic environment transitions. When transitions are stochastic, as is often the case in complex, real-world environments, the efficacy of the multi-turn DPO loss becomes unpredictable. It would be beneficial to see an analysis or empirical evidence that your approach is effective for the stochastic environment.

**Scalability and Computational Cost:**

The integration of MCTS, while enhancing performance, comes at a significant computational cost. For real-time applications or large-scale deployments, this could be prohibitive. A deeper analysis of the computational overhead and potential optimizations to mitigate this would be valuable. Could you elaborate on any strategies you've considered or implemented to address this?

**Limited Real-World Validation:**

Your experiments in WebShop and a real-world booking scenario are impressive, but they represent a limited scope of real-world environments. It's unclear how your framework would generalize to other domains or more complex applications. Broadening the range of scenarios tested would provide a more thorough validation of your framework's robustness and versatility.

**Sparse Reward Challenges:**

The sparsity of rewards in long-horizon tasks poses a significant challenge for your framework. While AI feedback and self-critique offer some mitigation, there's still a noticeable struggle with credit assignment. Exploring more sophisticated reward shaping or alternative methods to provide denser feedback could significantly enhance performance in these scenarios.

[1] Xiong, Wei, et al. "Building math agents with multi-turn iterative preference learning." *arXiv preprint arXiv:2409.02392* (2024).

**Questions:**

Please refer to the weaknesses.

---

> ### Author Response · Authors · 2024-12-02
> **Thank you!**
>
> We would like to thank the reviewers for the kind words and constructive feedback.
>
>
> ***”Concerns Regarding Multi-turn DPO Loss…”****
>
> It is indeed true that the standard DPO multi-turn loss depends on deterministic dynamics. In our own experiments, the OpenTable environment is non-deterministic (it’s a live website), while WebSHop is. We hypothesize that this is one of the reasons why outcome-level only DPO training performs relatively worse on OpenTable, while tree-based search data seems to offer larger improvement as it improves credit assignment.
> We should also note that under a slight change in perspective on preference modelling, the DPO loss is still valid in stochastic environments as well [1]. To what degree we can assume this holds in agentic settings is an open question.
>
>
> ***Scalability and Computational Cost:***
>
> We believe that computational costs can be manageable, while the biggest bottleneck for agents is the environment interaction. When the environment is “live” (such as a real website with real bookings) this can be quite costly and hard. While some approaches can be used to improve efficiency (such as our step-level critic) it is likely this is a necessary part for training advanced reasoning agents.
> We believe that effective approaches for this could mimic efforts in robotics, such as a mix of “sim” and real - i.e. containerized simulated environments and real data used together. The advantage of our approach is that the DPO training can be carried on off-policy (and even fully offline) data.
>
>
> ***Limited Real-World Validation***
>
> We agree that our application is somewhat limited, however training general-purpose web agents is still very much an open problem. Our work aims to be an initial step in that direction. In the meantime we have been working on extending our results to a wider set of simulated web applications and evaluating the potential sim-to-real results. This remains work in progress due to the complexities of accurate environment design.
>
>
> ***Sparse Reward Challenges***
>
> The reviewer is indeed correct. This was the main motivation behind our use of the step-level critic model, which does improve performance in credit assignment.  Ideally we would use a Process Reward Model (PRM) for this ranking as a prior over actions to explore first, however we believe training one would require a significant amount of either human data or Monte-Carlo rollouts, hence we opted for a zero-shot LLM-as-a-judge approach.
>
> [1]   Contrastive Preference Learning: Learning from Human Feedback without RL, Joey Hejna, Rafael Rafailov, Harshit Sikchi, Chelsea Finn, Scott Niekum, W. Bradley Knox, Dorsa Sadigh, 2024

---

> > ### Comment · Reviewer_LMY8 · 2024-12-02
> >
> > Thank you for your detailed response. I increase my confidence by 1.

---

### Official Review · Reviewer_Zp7s · 2024-11-01

**Soundness:** 2
**Presentation:** 1
**Contribution:** 3
**Rating:** 5
**Confidence:** 4

**Summary:**

This paper tackles the problem of building autonomous agents for web navigation tasks, including webshop and OpenTable. The proposed approach, AgentQ, leverages MCTS to generate an offline dataset and adopts DPO to fine-tune a base model based on both positive and negative trajectories. Finally, by combining AgentQ and online MTCS, the final performance substantially outperforms existing baselines.

**Strengths:**

1. The final web agent presented in this paper achieves state-of-the-art performances in particularly challenging environments. The result, I believe, is impressive.

2. The idea of combining search and RL to improve LLM's reasoning capabilities is novel. This project is one of the pioneer works that make this combination work. It would be inspiring for the community.

So, in principle, I believe this project is worth a presentation opportunity at top-tier ML conferences.

**Weaknesses:**

My biggest complaint about this paper is its writing. This paper is particularly poorly written. It is such a great project but presented as an arbitrary technical report. Let's take the STaR (https://arxiv.org/pdf/2405.00451) paper as a good writing reference and I will leave a few detailed comments below.

1. There is a significant lack of definitions. Throughout the paper, there is no definition of AgentQ. What do you mean by AgentQ? What is the difference between AgentQ and other baselines like DPO/STaR/XLaM? In the experiments, you use both AgentQ and AagentQ+MTCS as method names in the plot. What are the differences? It seems AgentQ itself requires MTCS for data generation. I spent a long time figuring out that the "MTCS" in Figure 1 refers to inference-time MTCS. Let's take the STaR paper as a reference. The STaR paper has a rigorous method section, where the definition of the method (including why it is named STaR) is clearly stated in Section 3. It even shows an overview plot in Figure 1 to make a reader understand the method better.

2. Many experiment and baseline details are lacking. In general, each paper should have a separate experiment section (like Section 4 in the STaR paper) to present the baselines, environment, and main results. However, in this paper, all the presentations are sequential without even a centralized experiment section, which brings substantial difficulties for readers to follow. It is also unclear to me how these baselines are selected. For example, why is there no RFT + MTCS demonstrated in Figure 1? Note that this baseline is listed in Figure 3. What are the current state-of-the-art numbers on these two benchmarks?

3. Lack of ablation studies. For example, for AgentQ, it is unclear to me how effective each component is. I think a separate ablation studies section should be presented. What if we run RFT with the same data collection pipeline? Does the same conclusion still hold for Webshop if we turn the AI-based action section off? What if we directly run the Q function as the final deployed policy without DPO training a policy? Moreover, since MTCS is involved, it is critical to me to see the performance growth w.r.t. the computation budget used by MTCS, both online and offline. I cannot see any detailed specifications of MTCS in the main paper. If we allocate an infinite amount of time, what's the limit of this approach? Any concrete case study?

4. The AI-based action selection protocol looks unclear to me. This protocol only ranks agents. But MTCS expands the actions using the UCB criteria. The data generation also depends on the Q function only. The ranking does not help learn the Q function. Why do you need an AI to rank the policies? My understanding of it is that it works as a branching mechanism to allow MTCS to only traverse those most promising actions. Is my understanding correct?

5. The theorem is good as it is but I don't think it makes too much sense for the overall project. You are not sampling data according to the  Q-induced distribution. The theorem and your algorithm aren't strongly connected. I would suggest you put more content on ablation studies and fine-grained controlled experiments.

6. Another minor point is that the citation formats are not consistent across the paper. Please update accordingly.

**Questions:**

See comments above.

In general, I would like to show my support for this project. It is a good one. However, the paper should be rewritten to get accepted.

I'm happy to raise my score after seeing the revised version.

---

> ### Author Response · Authors · 2024-12-02
>
> We would like to thank the reviewer for the kind words and suggestions! We have made many changes to incorporate the feedback in our manuscript.
>
> We agree that some definitions were unclear and have changed them. We have added more explicit definitions of AgentQ and AgentQ+MCTS so that there is no longer any ambiguity. We have also addressed many other writing improvements.
>
> ***In general, each paper should have a separate experiment section (like Section 4 in the STaR paper) to present the baselines…***
>
> We structured our presentation to present initial results and the motivation for the development of our method first. In particular, in standard RLHF, well-executed RL training matches performance of inference-optimization approaches [1]. This is the direction reflected in our “Preliminary Approach With Outcome Supervision” section. However, in our experiments, it emerged this is not the case in multi-turn agentic domains, where search proved to be critical, both for learning credit assignment as well as performance at deployment time. This observation leads to our next two sections.
>
> We have made some additional changes to the layout of the paper following your review. We have rewritten the discussion of the full WebShop results so that both the WebShop and OpenTable results are discussed in one experimental section. In addition, we have made the analysis more clear. This does improve the readability of the paper as now section 5 is focused on methods and section 6 contains results.
>
> ***Why is there no RFT + MTCS demonstrated in Figure 1? Note that this baseline is listed in Figure 3***
>
> In the WebShop domain, the performance between the base xLaM model and the RFT fine-tune is relatively close and they perform comparably with the addition of inference-time search. We use the RFT fine-tune on OpenTable since the base model is quite poor in comparison so it is not worthwhile to perform MCTS with it.
>
> ***What are the current state-of-the-art numbers on these two benchmarks?***
>
> As far as we are aware the current best results (success rates) on WebShop in a zero-shot approach are comparable to AgentQ zero-shot performance, and significantly lacking the performance with additional inference-time search.
>
> The OpenTable environment is not an existing benchmark, but a real-world website, which we used to demonstrate training and deployment in a real scenario. In our own evaluations the human-performance is comparable to AgentQ+MCTS with errors coming from task ambiguity and similar restaurant names with different locations.
>
> ***Lack of ablation studies…***
>
> We carry out large number of ablations on our OpenTable real environment training such as:
>
> Full-trajectory level outcome versus search-based training (DPO vs AgentQ).
> The effect of the step-level self-evaluation (AgentQNoAIFeedback)
> Zero-shot performance + Inference-time search (AgentQ vs AGentQ+MCTS)
>
> We did not include all of these in the WebShop experiment, since the environment is significantly simpler and does not require as extensive credit assignment as the more challenging OpenTable domain, hence the effect of these design choices is not as clear.
>
>
> ***What if we directly run the Q function as the final deployed policy without DPO training a policy?***
>
> We should note that we do not train a separate value function, but only construct MC estimates to guide our search. We believe that training an accurate value network would require a significant amount of rollouts and interactions, hence why we choose to use an LLM for zero-shot process guidance. This is essentially the “RFT + MCTS” baseline, which uses the base RFT policy and critic models without any additional training.
> ***If we allocate an infinite amount of time, what's the limit of this approach? Any concrete case study?***
>
> We ran the deployment-search until the model reached success (as judged by the critic) or hit an interaction limit. In this case, by design we found steady improvement with additional search ranging from the AgentQ zero-shot performance to AGentQ+MCTS level performance.

---

> > ### Author Response · Authors · 2024-12-02
> >
> > ***The AI-based action selection protocol looks unclear to me…***
> >
> > This is done to improve the efficiency of exploration and search by prioritizing more promising directions to explore. Ideally we would use a Process Reward Model (PRM) for this ranking as a prior over actions to explore first, however we believe training one would require a significant amount of either human data or Monte-Carlo rollouts [2]. Instead, we used an LLM-as-a-judge approach. We found that ranking actions seems to produce better step-level evaluations than individual point-wise scoring, although we did not run extensive ablations on this.
> >
> >
> > [1] AlpacaFarm: A Simulation Framework for Methods that Learn from Human Feedback
> > Yann Dubois, Xuechen Li, Rohan Taori, Tianyi Zhang, Ishaan Gulrajani, Jimmy Ba, Carlos Guestrin, Percy Liang, Tatsunori B. Hashimoto, 2023
> >
> > [2] Math-Shepherd: Verify and Reinforce LLMs Step-by-step without Human Annotations
> > Peiyi Wang, Lei Li, Zhihong Shao, R.X. Xu, Damai Dai, Yifei Li, Deli Chen, Y.Wu, Zhifang Sui, 2023

---

### Official Review · Reviewer_6cxt · 2024-11-03

**Soundness:** 4
**Presentation:** 2
**Contribution:** 2
**Rating:** 5
**Confidence:** 5

**Summary:**

This paper introduces AgentQ, an approach to enhance the capabilities of LLM agents in executing complex, multi-step planning and reasoning tasks within interactive environments. The integration of Monte Carlo Tree Search (MCTS) and iterative fine-tuning using step-level Direct Preference Optimization (DPO) has been demonstrated to be effective in LLM reasoning tasks. The authors apply the approach to both a simulated e-commerce task and the real-world booking scenario, incorporating several agent-related components. Empirical results demonstrate that AgentQ shows significant improvements over baseline methods and even surpassing average human performance under certain conditions.

**Strengths:**

The proposed method is successfully applied to the real-world booking scenario and achieve a $95.4\%$ success rate after a single day of data collection.

**Weaknesses:**

The experimental section is incomplete. The ablation study is entirely missing, providing no concrete indications of the strength of each component or design choice, such as the self-critique mechanism, the four types of actions, the effects of training sample sizes, and the weighted coefficient $\alpha$. While I understand the challenges of conducting experiments in real scenarios, ablations in the WebShop context are essential to validate the method's effectiveness. Furthermore, there is no analysis of the generated trajectories, nor are there case studies demonstrating the proposed method's efficacy.

The success rate of AgentQ+MCTS remains significantly lower than that of human experts. The authors attribute this discrepancy to short trajectories. More advanced LLMs should be evaluated to support this conclusion, such as LLaMA-3-70B-Instruct used in the booking scenario.

Many critical descriptions of the methods are lacking; please refer to the questions section.

Several important hyperparameters, such as $\theta_{threshold}$, are not listed.

The definition of $\tau$ in Equation 5 is missing.

**Questions:**

The authors state that the agent trained by DPO behaves greedily (Line 288). Does the step-level DPO address this issue?

Utilizing planning actions could deteriorate performance[1]. Did the authors observe similar outcomes?

How is the ranking of responses transferred to $\hat Q$?

How are the success rates of Human (average) and Human (expert) shown in Figure 1 obtained?

Which correspond to the explanation action in Figure 4?

References:
[1] Liu Z, Yao W, Zhang J, et al. Bolaa: Benchmarking and orchestrating llm-augmented autonomous agents[J]. arXiv preprint arXiv:2308.05960, 2023.

---

> ### Author Response · Authors · 2024-12-02
> **Thank you for your detailed feedback**
>
> ***The experimental section is incomplete….***
>
> The goal of our work is to develop pipelines for training and improved reasoning of LLM agents. As such, we do not aim to ablate the design choices of the core agent such as memory, action formulation, planning, prompting etc…
>
> We carry out large number of ablations on our OpenTable real environment training such as:
>
> 1. Full-trajectory level outcome versus search-based training (DPO vs AgentQ).
> 2. The effect of the step-level self-evaluation (AgentQNoAIFeedback)
> 3. Zero-shot performance + Inference-time search (AgentQ vs AGentQ+MCTS)
>
> We did not include all of these in the WebShop experiment, since the environment is significantly simpler and does not require as extensive credit assignment as the more challenging OpenTable domain, hence the effect of these design choices is not as clear.
>
> ***Many critical descriptions of the methods…***
>
> We would like to thank the reviewer for the detailed feedback on the presentation of our work and are working on incorporating their feedback.
>
> ***The authors state that the agent trained by DPO behaves greedily (Line 288). Does the step-level DPO address this issue?***
>
> No, we observe consistent model collapse in the agent behaviour with different RL techniques. While we can improve zero-shot performance, it still lags behind expert-level actions. This is largely resolved with additional inference-time search (AgentQ+MCTS). Since the release of our work similar effects have been observed in simpler domains such as math.
> We believe that using inference-time search techniques for agents is going to be a significant research and development direction over the next several years.
>
> ***Utilizing planning actions could deteriorate performance[1]. Did the authors observe similar outcomes?***
>
> We did not observe such an effect, however, we did not ablate the core agent design choices to a significant degree.
>
>
> ***How is the ranking of responses transferred to Q^ ?***
>
> We assign a normalized score to each action based on its ranking. We observed this to be more efficient than exploring actions based on individual pointwise scoring and evaluations.
>
>
> ***How are the success rates of Human (average) and Human (expert) shown in Figure 1 obtained?***
>
> These were obtained from the original work [1]. We should note that the challenge of the environment seems to originate from the ambiguity of product descriptions. In our own attempts, success rates varied between 40%-55% depending on the evaluator.
>
> ***Which correspond to the explanation action in Figure 4?***
>
> Figure 4 shows the input and output formats of the prompt for the base agent used across all experiments.
>
> [1] WebShop: Towards Scalable Real-World Web Interaction with Grounded Language Agents
> Shunyu Yao, Howard Chen, John Yang, Karthik Narasimhan, 2023

---

### Official Review · Reviewer_nCbR · 2024-11-04

**Soundness:** 2
**Presentation:** 3
**Contribution:** 1
**Rating:** 5
**Confidence:** 4

**Summary:**

The authors introduce an approach to train web agents, named *Agent Q*. It generates training data with MCTS over web pages and performs preference-based learning (DPO) over probable actions in each web page. As they assume sparse, trajectory-level success-or-failure feedbacks from the environment, the LLM-based critic is employed to generate such step-level feedback. The authors perform evaluation of the proposed approach on one existing benchmark, WebShop, and their own benchmark on OpenTable, and they demonstrate that their approach outperforms the employed baseline methods.

**Strengths:**

- The proposed approach is mostly sound (except for the open-loop planning component). MCTS can be an effective method for gathering trajectory data for agent training, and preference-based learning could be useful in web agent training scenarios.
- Overall, the manuscript is clear and easy to follow. The figures are made well for fair overviews. The writing is clear in general.

**Weaknesses:**

- The proposed approach generates a plan at the beginning of each trajectory, which seems to stay frozen for the rest. This open-loop planning can be a bottleneck for improving agents' capabilities in more general, complex scenarios.
- Despite the expressions such as "real-world booking scenarios" (Abstract) and "Scaling To Real World Websites" (Section 6), the proposed OpenTable experiment doesn't seem ideal for testing the proposed approach's "real-world" capabilities. The set of tasks are generated "by combining the restaurant name, desired date and time, and user information" (L1061), and the evaluation is done based on "(1) date and time set correctly, (2) party size set correctly, (3) user information entered correctly, and (4) clicked complete reservation" (L1053). These imply that the training and test tasks (and thus correct trajectories) all share similar structures with the same set of constraints, and it can make it hard to assess the generalization capabilities of the approach, which is an important part of being "real-world." Besides, the proposed approach uses a frozen plan for each trajectory, which can be a suitable design for OpenTable but not for most of real-world websites and scenarios.
- According to the WebShop results, the advantage of the proposed approach, especially over the trajectory-level DPO baseline doesn't seem very clear to me (40.6% vs 41.5% from Figure 1).
- Combining MCTS, self-evaluation, and step-level preference learning for training has already been explored in prior work (e.g., Xie et al., 2024, as mentioned in this submission).

**Questions:**

Please check out the weaknesses section.

---

> ### Author Response · Authors · 2024-12-02
>
> We would like to thank the reviewer for the detailed feedback!
>
> ***The proposed approach generates a plan at the beginning of each trajectory, which seems to stay frozen for the rest. This open-loop planning can be a bottleneck for improving agents' capabilities in more general, complex scenarios.***
>
> The plan is generated as part of the first agent step and thus in MCTS we can backtrack (and occasionally do) to the initial stage and re-generate a new plan.
>
>
> ***Despite the expressions such as "real-world booking scenarios" (Abstract) and "Scaling To Real World Websites" (Section 6)...***
>
> We do agree that our proposed tasks are somewhat narrow in their formulation, however training general-purpose agents in real domains is still a major unsolved problem. We believe the major challenge in this domain is the dynamic, non-stationary and real-time environment of a live website as availability and dining options change minute-to-minute.
>
> This is in stark contrast to prior reasoning works which combine tree search with LLMs and RL, as they operate in static MDPs, such as solving high-school math. As far as we are aware our work is the first one to show such capabilities in non-deterministic non-static domains.
>
>
> ***According to the WebShop results, the advantage of the proposed approach, especially over the trajectory-level DPO baseline doesn't seem very clear to me (40.6% vs 41.5% from Figure 1).***
>
> This is indeed correct, but the result is based on the more simplistic nature of the WebShop environment. The core idea of our tree-search based training is to help the agent learn credit assignment in complex multi-step (and potentially dynamic) environments.
>
> In this case we use WebShop as a development proof-of-concept environment. This domain is deterministic and relatively simple as it only requires about six steps to solve. Hence here credit assignment is not as much of an issue.
>
> In contrast, the real OpenTable environment requires about 13 steps to solve and is non-deterministic, thus credit assignment issues become more prominent and the gap between purely outcome-driven training versus the tree-structured exploration we use becomes more apparent.
>
>
>  ***Combining MCTS, self-evaluation, and step-level preference learning for training has already been explored in prior work (e.g., Xie et al., 2024, as mentioned in this submission).***
>
> The main contribution of our work is to extend approaches based on search and LLMs from relatively simpler static domains such as math to realistic and more complex agentic dynamic scenarios, which presents unique challenges not previously explored as far as we’re aware.

---

> > ### Comment · Reviewer_nCbR · 2024-12-03
> > **Response to Authors**
> >
> > Thank you for the clarification about the planning. However, tree expansion can happen after the initial step, and I still think the initial planning part of the method needs more justification (which can be done with empirical evaluations).
> >
> > While I do appreciate the authors' response to my review in addressing my concerns, after reviewing their response, I find that my original evaluation on this work remains unchanged.

---

### Official Review · Reviewer_LfUP · 2024-11-04

**Soundness:** 2
**Presentation:** 1
**Contribution:** 2
**Rating:** 5
**Confidence:** 4

**Summary:**

The paper proposes Agent Q, an online fine-tuning framework for enhancing reasoning capabilities in autonomous AI agents for web navigation tasks. Agent Q combines DPO, MCTS, and process supervision to improve the agent's performance. The experimental results show that Agent Q outperforms the baselines on two benchmark tasks, highlighting its potential for real-world applications.

**Strengths:**

- The paper proposes a method for online fine-tuning a web navigation agent and the empirical results show that the proposed method outperforms the baseline methods.
- The paper evaluates Agent Q on both simulated and real-world tasks, demonstrating its effectiveness in various scenarios.

**Weaknesses:**

- The idea is not novel enough since Agent Q uses a combination of DPO, MCTS, and process supervision for web navigation task.
- Lack of experimental details, such as hyperparameters, and the significance of the results. Also, the code is not provided.
- Lack of ablation studies to analyze the effects of number of iterations in Agent Q.

Format and writing issues:

- The paper cites papers using wrong LaTeX commands. For example, "(Zhou et al., 2024c) has shown success
formulating the RL problem at a step level" should be changed into "Zhou et al. (2024c) has shown success
formulating the RL problem at a step level" by using the `\citet` command. "Classical RLHF has used policy gradient type of algorithms, such as PPO Schulman et al. (2017), ... " should be changed into "Classical RLHF has used policy gradient type of algorithms, such as PPO (Schulman et al., 2017), ... " by using the `\citep` command. I strongly recommend the authors to use the correct LaTeX commands for citations.
- "will produce actions $\mathbf{a}_t \sim \pi \mathbf{a} | \mathbf{h}_t)$" should be "will produce actions $\mathbf{a}_t \sim \pi(\mathbf{a} | \mathbf{h}_t)$". The same issue in Eq. (2).
- Inconsistent equation referencing: In Algorithm 1, the equation is referenced as (9) and Eq. (4), but in the text, it is referenced as Eq. 2. The same issue for figures. There are "Figure 1" and "Fig. 1" in the text. I recommend the authors to check the consistency of the referencing.
- Replicated references in the paper, such as "Appagent: Multimodal agents as smartphone users", "Chain of preference optimization: Improving chain-of-thought reasoning in llms".
- "In this environment, training with outcome-supervision only DPO further improves performance by 4.6% to 71.8% but significantly under-performs the full Agent Q pipeline which achieves a zero-shot success rate of 81.7%" does not have a period at the end of the sentence. I recommend the authors to check the punctuation of the sentences in the paper.

**Questions:**

Besides the weaknesses above, further questions are as follows:

- "RFT and DPO over xLAM-v0.1-r demonstrate improvements in performance from 28.6% to 31.3% and 37.5% respectively." However, 37.5% is the result of GPT-4 in Figure 1, not DPO. So what are the correct results of GPT-4 and DPO?
- "We use the base model to produce a feedback score for each action by asking it to rank the generated actions by its perceived utility in helping the agent complete the user task." Can the authors provide more details on how the feedback score is generated?
- How does the number of iterations in Agent Q affect the performance? Are there any ablation studies on this?
- How does the computational complexity of Agent Q compare to the baseline methods?

---

> ### Author Response · Authors · 2024-12-02
> **Thank you for your review**
>
> We would like to thank the reviewer for the detailed feedback.
>
> ***“The idea is not novel enough since Agent Q uses a combination of DPO, MCTS, and process supervision for web navigation task.”***
>
> The general research direction of combining LLMs with search capabilities is relatively new with most works focusing on model-training in the math domain developed within the last 6-9 months. We believe that our work is the first one to scale these ideas to realistic agentic applications beyond simple tasks such as the Game of 24 or high-school math. Moreover, we believe that advanced reasoning capabilities for agentic models will be the main research and development direction in artificial intelligence over the next two years and our work is an early attempt in that direction.
>
> As for DPO, MCTS, and process supervision, the relevant works have been referenced and contextualized in the paper.
>
> ***Lack of experimental details, such as hyperparameters, and the significance of the results. Also, the code is not provided.***
>
> Due to the large-scale interaction requirements with real websites as well as training costs, we cannot provide significance over multiple runs, but this has been standard in LLM training literature since the rise of “large” foundational models.
>
> Unfortunately we cannot open-source our code, but we still believe that our research and results will be informative and useful to the community.
>
>
> ***Lack of ablation studies to analyze the effects of number of iterations in Agent Q.***
> All reported results are over 3 iterations. While the RFT model continues improving, the DPO-based models largely saturate in performance by iteration 3. This is consistent with observations on efficiency in simpler domains, such as math [1]
>
>
>
> On the presentation, we have incorporated all your suggestions into the updated draft. Thank you for the effort and detail you provided in your feedback. It has helped us to improve the quality of the paper.
>
> ***"RFT and DPO over xLAM-v0.1-r demonstrate improvements in performance from 28.6% to 31.3% and 37.5% respectively." However, 37.5% is the result of GPT-4 in Figure 1, not DPO. So what are the correct results of GPT-4 and DPO?***
>
> We apologize for this typo, Figure 1 has the correct performance numbers and we have fixed the references to it.
>
> ***How does the computational complexity of Agent Q compare to the baseline methods?***
>
> While our agentic pipeline uses multiple components the core costs are still based on model processing of inputs and generating outputs. Thus the computational requirements for training are somewhat comparable (not accounting for additional search at agent deployment time).
>
>
>
> [1] RL on Incorrect Synthetic Data Scales the Efficiency of LLM Math Reasoning by Eight-Fold, Amrith Setlur, Saurabh Garg, Xinyang Geng, Naman Garg, Virginia Smith, Aviral Kumar, 2024

---

### Meta-Review · Area_Chair_pmHm · 2024-12-21

**Metareview:**

This paper proposes an online framework to improve agents' autonomous reasoning capabilities in web navigation. There are two main concerns. First, the method and experiments (including ablation studies) lack details, which make it hard to follow the manuscript. Second, the novelty of the method. DPO, MCTS and the combination of those have been proposed by prior work. However, web navigation can be a new downstream task, which has its own unique challenges. In this regard, dismissing a paper's contribution purely because of its use of prior work may not be justified. However, a more detailed and careful discussion and validation of why using those methods and the combination of those method can be a good idea to address the challenges in web navigation( and what is the unique aspect of web navigation) is needed for readers to better appreciate this work.

**Additional Comments On Reviewer Discussion:**

Unfortunately, there is not much going back and forth during the discussion period. However, after checking the rebuttal and reviewers' comments, the sharing concern regarding the details of the method, experiments and analysis of ablation studies remains.

---

### Decision · Program_Chairs · 2025-01-22

Reject